# Critical evaluation of an autologous peripheral blood mononuclear cell-based humanized cancer model

**Harinarayanan Janakiraman**[1], **Scott A. Becker**[2], **Alexandra Bradshaw**[3], **Mark P. Rubinstein**[4], **Ernest Ramsay Camp**[1,5,6]*

**1** Michael E. DeBakey Department of Surgery, Baylor College of Medicine, Houston, Texas, United States of America, **2** Molecular and Systems Pharmacology, Emory University, Atlanta, GA, United States of America, **3** Department of Surgery, Medical University Of South Carolina, Charleston, SC, United States of America, **4** The Pelotonia Institute for Immuno-Oncology, Ohio State University Comprehensive Cancer Center–James, Columbus, OH, United States of America, **5** Dan L. Duncan Comprehensive Cancer Center, Houston, Texas, United States of America, **6** Michael E. DeBakey VA Medical Center, Houston, Texas, United States of America

* ramsay.camp@bcm.edu

**Data Availability Statement:** All relevant data are within the paper and its Supporting Information files.

## Abstract

The use of humanized mouse models for oncology is rapidly expanding. Autologous patient-derived systems are particularly attractive as they can model the human cancer's heterogeneity and immune microenvironment. In this study, we developed an autologous humanized mouse cancer model by engrafting NSG mice with patient-derived xenografts and infused matched peripheral blood mononuclear cells (PBMCs). We first defined the time course of xenogeneic graft-versus-host-disease (xGVHD) and determined that only minimal xGVHD was observed for up to 8 weeks. Next, colorectal and pancreatic cancer patient-derived xenograft bearing NSG mice were infused with 5x10$^6$ human PBMCS for development of the humanized cancer models (iPDX). Early after infusion of human PBMCs, iPDX mice demonstrated engraftment of human CD4+ and CD8+ T cells in the blood of both colorectal and pancreatic cancer patient-derived models that persisted for up to 8 weeks. At the end of the experiment, iPDX xenografts maintained the features of the primary human tumor including tumor grade and cell type. The iPDX tumors demonstrated infiltration of human CD3+ cells with high PD-1 expression although we observed significant intra and inter-model variability. In summary, the iPDX models reproduced key features of the corresponding human tumor. The observed variability and high PD-1 expression are important considerations that need to be addressed in order to develop a reproducible model system.

## Background

The advent of immune checkpoint therapy (anti-PD-1(L)1 monoclonal antibodies (mAbs)) has heralded great success in a growing number of cancers such as melanoma [1]. Unfortunately, almost no pancreatic cancer and few colorectal cancer patients respond to immune checkpoint therapy [2]. The addition of cytotoxic chemotherapy to immune checkpoint

**Funding:** This work was supported in part by funding from MERIT REVIEW AWARD (Issue date: 1/15/2019) Department of Veteran Affairs and Clinical Science Research and Development (1I01CX001880-01A1) for the research project titled "Targeting Sphingosine-1-phosphate to overcome SNAI1-mediated therapy" issued to C.E. R. Supported in part by the Translational Science Shared Resource, Hollings Cancer Center, Medical University of South Carolina (P30 CA138313). Supported in part by the Biorepository & Tissue Analysis Shared Resource, Hollings Cancer Center, Medical University of South Carolina. The funders had no role in study design, data collection and analysis, decision to publish, or preparation of the manuscript.

**Competing interests:** The authors have declared that no competing interests exist.

therapy offers great potential for overcoming resistance mechanisms, promoting lymphocyte infiltration into the tumor, and improving clinical responses. Landmark studies in NSCLC and breast cancer suggest favorable interaction with platinum-based chemotherapies and anti-PD-1 mAb [3,4]. New clinical approaches still desperately need optimization for important variables such as agent order, timing, and dose. While preclinical mouse syngeneic cancer and genetically engineered mouse models are invaluable [5,6], the research community still lacks platforms that can faithfully reproduce the patient's cancer heterogeneity and the tumor-immune interaction.

Patient-derived xenograft (PDX) models have generated enthusiasm as they preserve human tumor heterogeneity and morphologic/genomic integrity [7–11]. Our group previously designed a highly translational PDX platform to study radiation therapy resistance and test novel therapeutic strategies for colorectal cancer [12,13]. However, traditional PDX models cannot account for the tumor-immune axis. "Humanizing" strategies including engraftment of human hematopoietic stem and progenitor cells (HSPC), human tumor infiltrating lympho-cytes (TILs) and human peripheral blood mononuclear cells (PBMCs) have been developed in severely immunodeficient mice such as the NSG (NOD/SCID-IL2γ-/-) mouse strain that lacks T, B, NK, and NKT cells [14–19]. Although the use of CD34+ HSPCs for cancer models is promising, HSPC-based models suffer from both biologic and logistic limitations [16–19]. Per-haps most important, the requirement for a bone marrow biopsy to obtain autologous HSPCs limits the feasibility of this humanizing approach for solid cancers such as PDAC [16,19]. Autologous humanized PDX models offer the possibility of recapitulating the patient's tumor-immune axis and allowing co-localization of human tumor-reactive T cells with MHC-matched tumors. While autologous models using either human PBMCs or TILs in xenograft models have been reported, such approaches still require standardization and validation before widespread use [15,20–22]. PBMC based models are appealing as the immune cell resource is readily available and accessible without any additional invasive procedures. However, this strategy is limited due to xenogeneic graft versus host disease (xGVHD). The timing of GVHD is dependent on many experimental parameters such as the use of sublethal radiation and immune cell products. Although reports have used this strategy to test therapy [22,23] the opti-mal timing and experimental parameters have not been established for cancer models.

Our goal was to establish an autologous humanized PDX model for reproducible immuno-therapy investigation. Our study determined technical aspects required to establish an autolo-gous humanized cancer model system and demonstrated the ability to successfully engraft patient PBMCs that infiltrate the xenograft. However, we encountered challenges that limited the application of this strategy for reproducible experimentation and for testing immunother-apy in a rigorous fashion. Here, we report our technical model approach and critical assess-ment of an autologous humanized cancer platform established with matched patient derived PBMCs and tumor xenografts.

## Materials and methods

### Patient tissue and xenograft processing

Patient tumor samples and blood were obtained from consenting colorectal and pancreatic cancer patients at the Medical University of South Carolina (MUSC) based on an Institutional Review Board (IRB) approved protocol. This study was also approved by the Institutional Ani-mal Care and Use committee (IACUC) at MUSC. Patient tumor tissues were surgically har-vested and placed immediately in HBSS with 1% penicillin-streptomycin buffer. A small portion of the patient's tumor tissues was fixed in 10% neutral buffered formalin (NBF) for overnight and rinsed in PBS briefly followed by incubation in 70% ethanol until further

analysis. The remaining tumor was then crushed with a syringe handle followed by blade cutting into <1mm$^3$ tumor fragments. After being washed with HBSS, tumor samples were resuspended in Ammonium-Chloride-Potassium (ACK) Lysing buffer (Fisher Scientific, Hampton, NH) to eliminate blood cells before being digested by 0.2U/mL of Liberase DH (Roche) containing 10um Y-27632 (Sigma Aldrich) in 8mL of HBSS with 1% penicillin-streptomycin for 1 hour at 37˚C. The cell suspension was filtered through a 250μm sieve and 100μm cell strainer (Fisher Scientific, Waltham, MA). The cell suspension was then collected and spun at 1000 rpm for 5 minutes at room temperature and then the cell pellet was resuspended in basal culture medium i.e. advanced DMEM/F12 supplemented with 10mM HEPES, 1x GlutaMAX and 1x penicillin/streptomycin (Invitrogen, Carlsbad, CA). After cell counting, tumor cells were re-suspended in 100% Matrigel (BD Biosciences) at 0.5 x 10$^6$ cells per subcutaneous injection into NSG mice to make initial PDX tumors (passage 0; P0). To make subsequent passaged PDX including first passage (P1) and second passage (P2) xenografts, the same protocol was used as we previously published [30]. At time of surgical resection of the tumor, up to 50mL of blood was collected in heparinized blood collection tubes from the same patients from whom the tumor was obtained. The tumor samples were stored in ice cold HBSS with 1% penicillin-streptomycin until further processing. Blood samples were processed using SepMate tubes and Lymphoprep density gradient solution (Stem cell technologies). hPBMCs were frozen in heat inactivated FBS with 10% DMSO at 20–30 x 10$^6$ cells per vial.

## Assessment of xGVHD in NSG mice

Briefly, 5 million hPBMCs were injected (i.v) into 6–8 weeks old NSG mice. After transplantation of hPBMCs, mice were ear punched and individual weights were obtained and recorded on +1 and weekly thereafter for 12 weeks or until the mice were euthanized. xGVHD was measured weekly based on 5 parameters: weight loss, posture, activity, fur, and skin over a 12-week period [24,25]. At the time of analysis, mice were evaluated and graded from 0 to 2 for each of the 5 parameters as mentioned above. To assess the severity of xGVHD, percentage of weight change in each mouse was measured. A threshold of 10% weight loss was used as a criterion to signify the presence of moderate GVHD. A clinical index with a maximum score of 10 was subsequently generated as previously described [24]. When mice developed hunched posture, showed severe weight loss, reduced movement, dehydration, or moribund, they were euthanized (CO2 inhalation followed by cervical dislocation).

## Development of humanized mouse model (iPDX)

After establishing P2 PDX xenografts and allowing subcutaneous tumors to grow to a volume of 100mm3 (3–9 weeks), patient matched PBMCs were thawed and then engrafted into the mice under anesthesia (Inhalation anesthesia with 2% Isoflurane). Each mouse was injected i.v. with 5 x 10$^6$ PBMCs in 100uL DPBS. Control mice received PBS alone. Per patient model, we established four experimental iPDX mice and four control PDX. Each iPDX mouse received cancer cells and PBMC from a single patient. Mouse weight was measured weekly, and engraftment was confirmed by flow cytometry two weeks following injection of PBMC (S1 Fig). Xenografts were established prior to PBMC infusion in order to maximize the time for tumor development prior to xGVHD. All efforts were made to minimize the suffering of mice.

## Flow cytometry analysis

At the indicated time points after transplantation of human PBMCs, peripheral blood was collected from mice with a lancet (Golden rod; Medipoint, INC., Mineola, NY). Additionally, at the time of necropsy, peripheral blood and tumors were harvested. Peripheral blood was depleted of

erythrocytes using ACK lysis buffer according to manufacturer's instructions. PBMCs were obtained by centrifuging at 1400 rpm for 5 min at room temperature. PBMCs were washed once with 1ml PBS followed by 1ml FACS buffer (DPBS+2% FBS). Tumor cells were obtained by crushing the tumors and digested with 5ml of RPMI 1640 (Gibco) containing 1mg/ml Collagenase D (Wako Chemicals). Digestion was stopped by adding RPMI complete medium. After centrifugation, the tumor cell suspension was resuspended in 15ml DPBS and filtered using 100uM mesh (Fisher Scientific, Hampton, NH). Cells ($1.5–2 \times 10^6$ cells/sample) were then stained with Live/Dead fixable stain kit (LDBUV395) (Invitrogen). The following antibodies specific for human antigens were used for flow cytometry staining: anti-CD3 (clone SK7), anti-CD4 (clone RPTA-4), anti-CD8 (clone SK1), anti-CD45, anti-PD-1 (clone EH12.1) (eBioscience). Cells were re-suspended in 50 μl FACS buffer and incubated with surface antibodies for 20 min at 4˚C in the dark and washed twice with FACS buffer. Cells were then fixed for 20 min on ice. Samples were analyzed by flow cytometry (LSR Fortessa, BD Biosciences) and Flowjo Software.

## Immunohistochemistry analysis

After performing euthanasia, tumors harvested from mice were fixed in neutral-buffered formalin for 24 hours and embedded in paraffin using standard techniques. 5um thick sections from the pre-treatment human tumor biopsy and xenograft samples were deparaffinized and stained for hematoxylin and eosin (H&E) as well as stained using the Ventana Discovery Ultra Automated Research Stainer (Roche Diagnostics Corp. Indianapolis, IN). Heat-induced epitope retrieval (HIER) was performed in EDTA buffer pH 9 (Cat.#S2367 Agilent/Dako Santa Clara, CA) for 32 minutes at 95˚C and endogenous peroxidase was blocked with a solution of hydrogen peroxide after incubation of the first primary antibody. Antibodies used included CD3 (Cell Signaling Technologies, clone E4T1B, 1:100) and PD-1 (AbCam, clone EPR4877(2), 1:100). After incubation with primary and secondary antibodies, DISCOVERY Purple and DISCOVERY ChromoMap DAB Kits (Roche Diagnostics Corp. Indianapolis, IN) were used for chromogen detection. Hematoxylin was used for counterstaining. Stained slides are mounted with Thermo Scientific™ Richard-Allan Scientific™ Cytoseal™ XYL (Thermo Fisher Scientific, Waltham, MA) and imaged using the Akoya Vectra® Polaris™ Automated Quantitative Pathology Imaging system (Akoya Biosciences, Marlborough, MA). Whole slide scans were done at 20X magnification and subsequently regions of interest (ROIs) where chosen at random across each tumor for further analysis. Spectral unmixing and basic cell phenotyping was performed using the inForm® Software v2.4.10 (Akoya Biosciences, Marlborough, MA) and resulting images were exported in TIFF format for further analysis. Spatial analysis and cell counting was performed using the PhenoptrReports Open-Source R Package (https://akoyabio.github.io/phenoptrReports/index.html, Akoya Biosciences, Marlborough, MA) Hematoxylin and eosin (H&E) slides were assessed for tumor type/differentiation and for cellularity by our surgical pathologist.

## Statistical analysis

Unless otherwise stated, statistical analyses were performed using the Student's t-test for paired data. P<0.05 was considered significant.

# Results

## Low dose of human PBMC induces xGVHD in NSG mice

Previous studies have demonstrated that infusion of PBMCs into NSG mice results in xGVHD, although the degree of xGVHD depended on experimental design parameters such

as the use of total body low-dose radiation and chemotherapy [14,22,24,25]. To design a protocol that optimized engraftment of human PBMCs while delaying xGVHD, we omitted low-dose radiation and immune ablative chemotherapy. Therefore, we injected (i.v) 5 million PBMCs into 6–8 weeks old NSG mice and then closely observed mice for signs of xGVHD over a 12-week period. We used 6–8 week old NSG for the assessment of xGVHD which have been shown to be less prone to severe xGVHD compared to older mice (12 weeks and above) [26,27]. As shown in S2A Fig, only 3 out 9 mice (33.3%) injected with 5 million PBMCs developed signs of moderate to severe xGVHD (GVHD Score >5) over a 12-week period. Eight weeks following infusion of PBMCs, there was only minimal evidence of xGVHD (average GVHD score = 2) in the entire group. Signs and symptoms of xGVHD rapidly progressed after 8 weeks. Six mice (66%) demonstrated mild signs of xGVHD (GVHD Score < 4) over the same time period (Supporting information, S2A Fig). All 6 mice with mild xGVHD had no weight loss and survived past 12 weeks. Over a period of 12 weeks, the cohort of mice continued to gain weight even as the signs of xGVHD appeared after 6 weeks and progressed up until 12 weeks (S2B and S2C Fig). Only one mouse developed severe xGVHD and was euthanized early. Taken together, the infusion of PBMCs into NSG mice leads to only minimal xGVHD for up to 8 weeks. Thus, we concluded that this experimental design could be used for modeling the human tumor-immune interaction and for therapy experiments lasting less than 8 weeks before being confounded by xGVHD.

## Human T-cells effectively engraft in iPDX mouse peripheral blood

We established iPDX models from one colon cancer (MCC) and one pancreatic cancer (MPC) patient to define experimental parameters. After xenograft tumors (s.c.) grew to 100mm$^3$, human PBMCs were injected by tail vein. Two weeks after PBMC infusion, a significant population of human CD3+CD45+ T cells was identified in the MCC iPDX blood samples relative to blood from PDX (no infused hPBMCs) (Fig 1A and 1B). In the MCC model, both CD4+ T cells (33.7 ± 12.2%) and CD8+ T cells (54.1 ± 16.4%) populations were identified in the iPDX peripheral blood (Fig 1C). Similar to the MCC model, human CD3+CD45+ T cells were detected in the MPC model peripheral blood sample 2 weeks after infusion although the population of engrafted CD3+ T cells was much less than the MCC model (5.42 vs 15.76%) (S3A Fig). In addition, the CD8 and CD4 ratio in the peripheral blood varied between patient-derived models (MCC 1.6:1) and MPC (1:35) at 2 weeks following infusion. Interestingly, a population of human T cells was identified in one MPC PDX mouse which was not adoptively transferred with human T cells (S3A Fig). This likely resulted from carryover of T cells during generation of the PDX as reported by others [21]. In this mouse, the CD3+ cells were only observed in the peripheral blood, did not expand over time, and were not present in the xenografts at the end of the experiment.

To understand the change in the engrafted human lymphocyte population over time, blood samples from iPDX mice were evaluated in a similar fashion at the end of the experiment at 8 weeks following infusion of PBMCs. In the MCC model, the population of engrafted CD3 +CD45+ T cells demonstrated minimal expansion from week 2 to 8 (Fig 1D, left). However, the engrafted CD3+CD45+ T cells population in individual iPDX mice demonstrated significant variability within the patient model (intra model variability) with a range of percentages from 6.78 to 29.7% at 2 weeks and 13.7 to 22.9% at 8 weeks. The engrafted fraction of both CD4+ and CD8+ T cells remained essentially constant from two to eight weeks after infusion of cells with a 1.6:1.4 ratio of CD8+ and CD4+ T cells (Fig 1D middle and right panel). In the MPC model, however, the population of CD3+CD45+ T cells significantly expanded over time. (S3D Fig, left panel). In contrast to the MCC model, the engrafted fraction of CD8+ T

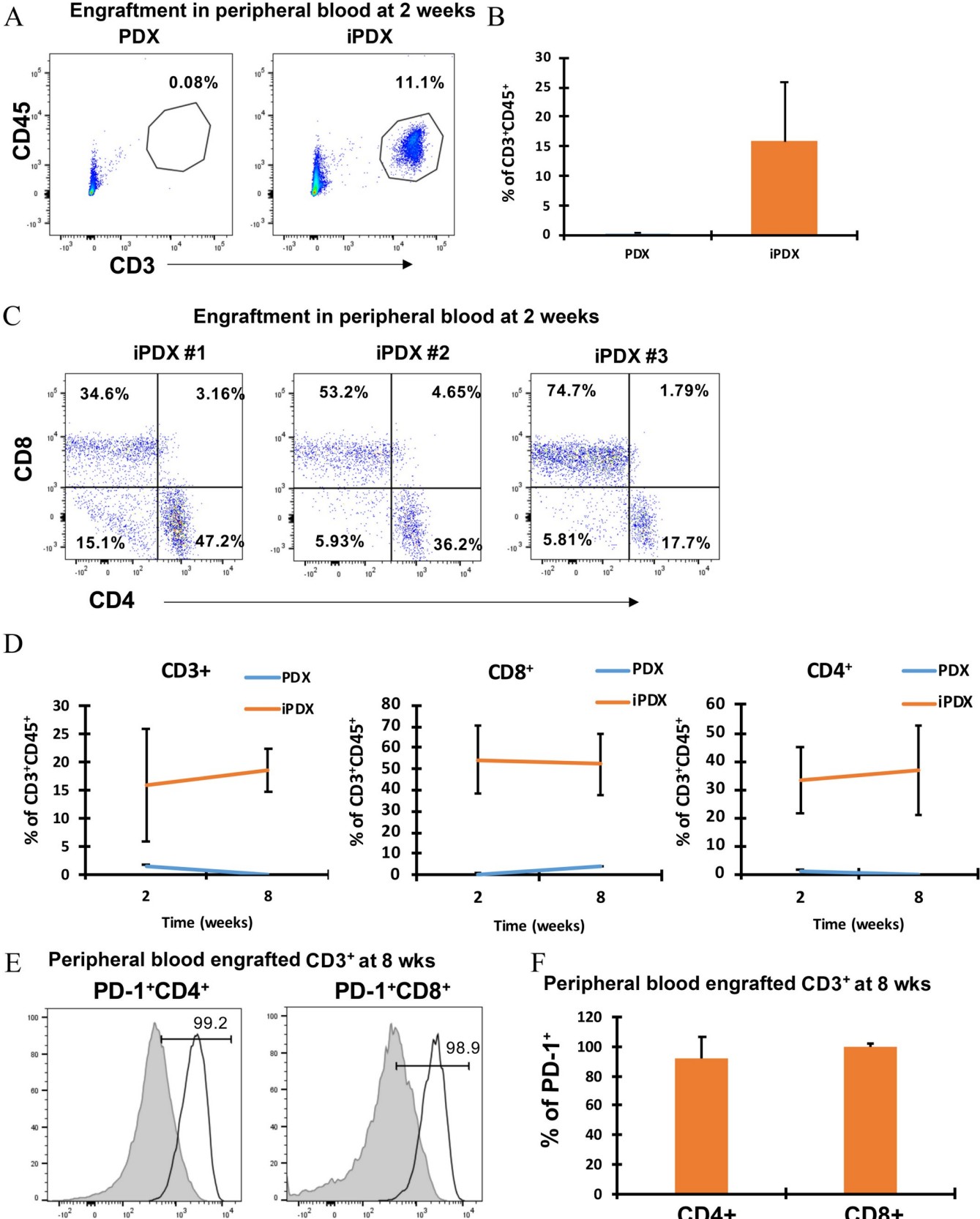

**Fig 1. Effective engraftment of human PBMC derived T cells in MCC iPDX mouse peripheral blood. (A)** Representative flow cytometry plots are shown for MCC model CD3+CD45+ T lymphocytes from PDX mice (left), iPDX mice (middle). **(B)** Bar graph of the average % of CD3+CD45+ cells (right). Data from peripheral blood samples collected at 2 weeks. **(C)** Flow cytometry plots for CD4+ and CD8+ T lymphocytes from all iPDX mice (n = 3). Data from peripheral blood samples collected at 2 weeks. **(D)** The change in the average % of human T lymphocytes from early (2 week) to late (7 weeks) engraftment are shown for MCC CD3+CD45+ T lymphocytes (left), CD8+ T lymphocytes (middle) and CD4+ T lymphocytes (right). **(E)** PD-1+CD4+ T lymphocytes (left) PD-1+CD8+ T lymphocytes (right). T lymphocytes from healthy human donor PB (grey histogram, left) and iPDX PB (empty histogram, right). **(F)** Bar graph showing average percentage of PD-1+CD4+ and CD8+ T lymphocytes. Data from peripheral blood samples collected at 8 weeks. The values represent the percentage of human CD3, CD4 and CD8 population in iPDX mice peripheral blood. Average of per group (n = 3) is shown.

cells in the MPC iPDX peripheral blood demonstrated a significant increase from early to late engraftment with a 1:75 ratio of CD8+ and CD4+ T cells at 2 weeks to 2:1 ratio at 8 weeks (S3D Fig, middle and right panel). For both models, a high percentage of engrafted peripheral CD4+ and CD8+ T cells demonstrated expression of PD-1 compared to a control human donor sample (Fig 1E and 1F, S3E and S3F Fig). In both patient-derived models, human CD4 + and CD8+ T cells successfully engrafted the NSG mice however, we observed significant intra- and inter-model variability. Taken together, our data demonstrate that infusion of cancer patient's PBMC samples into NSG mice implanted with matched PDX reproducibly engrafts human CD4+ and CD8+ T cells that can persist up to 8 weeks however, the engrafted human T cells demonstrated significant intra- and inter-model variability.

## Tumor xenografts from iPDX mice demonstrate cellular and architectural heterogeneity reproducing key features of the human cancer

Histopathologic comparison of the human tumor and corresponding early xenograft (P2) by a gastrointestinal pathologist demonstrated concordance for tumor cellularity, differentiation, and morphology for the human cancer, PDX and iPDX in both the MCC and MPC models (Fig 2A and 2B, top panel).

To assess the tumor infiltrating lymphocytes, we first analyzed expression of human CD3 in the human tumor as well as the corresponding iPDX and PDX. Xenograft tumors from the iPDX demonstrated a significant intra-tumoral population of human CD3+ T cells while, as expected, tumor from PDX mice were essentially void of CD3+ T cells (Fig 2A and 2B, middle panel). Moreover, the human CD3+ T cells were detected in the stromal and intraepithelial regions of both iPDX and patient tumor in the MCC and MPC models. In both iPDX models, the quantitated levels of human CD3+ expression in the iPDX tumor was higher than in the corresponding human tumor (Fig 2C and 2D). In contrast to the human tumors, the majority of human CD3+ cells from the iPDX tumors demonstrated detectable PD-1 expression by IHC (Fig 2A and 2B bottom panel, 2C and 2D).

Flow cytometry analysis of the immune cell populations in the xenografts confirmed the IHC findings. In both patient-derived models, a significant increase in human CD3+CD45+ T cell population was detected in the xenografts (Fig 3A and 3B, Supporting information, S4A and S4B Fig). Phenotypic analysis of human CD3+ T cells from the MCC iPDX xenografts identified both CD4+ T cell (31.1 ± 21%) and CD8+ T cell (52.6 ± 12.6%) populations (Fig 3C). Similarly, both populations were detected in the MPC iPDX xenografts (Supporting information, S4C Fig). Large variations were identified in the infiltrating cell populations in both the MCC and MPC iPDX mice. Similar to the IHC analysis, xenograft infiltrated CD4+ and CD8+ T cells from both iPDX models expressed high levels of PD-1 (Fig 3D and 3E, Supporting information, S4D and S4E Fig). In contrast, the percentage of CD4+ and CD8+ T cells from our PDX control mice were undetectable (Fig 3B). Taken together, these results suggest that transfer of human PBMC resulted in xenograft infiltration of human T cells however, the infiltrating T cell population was significantly increased compared with the corresponding human tumor with a significantly higher population of PD-1 positive cells.

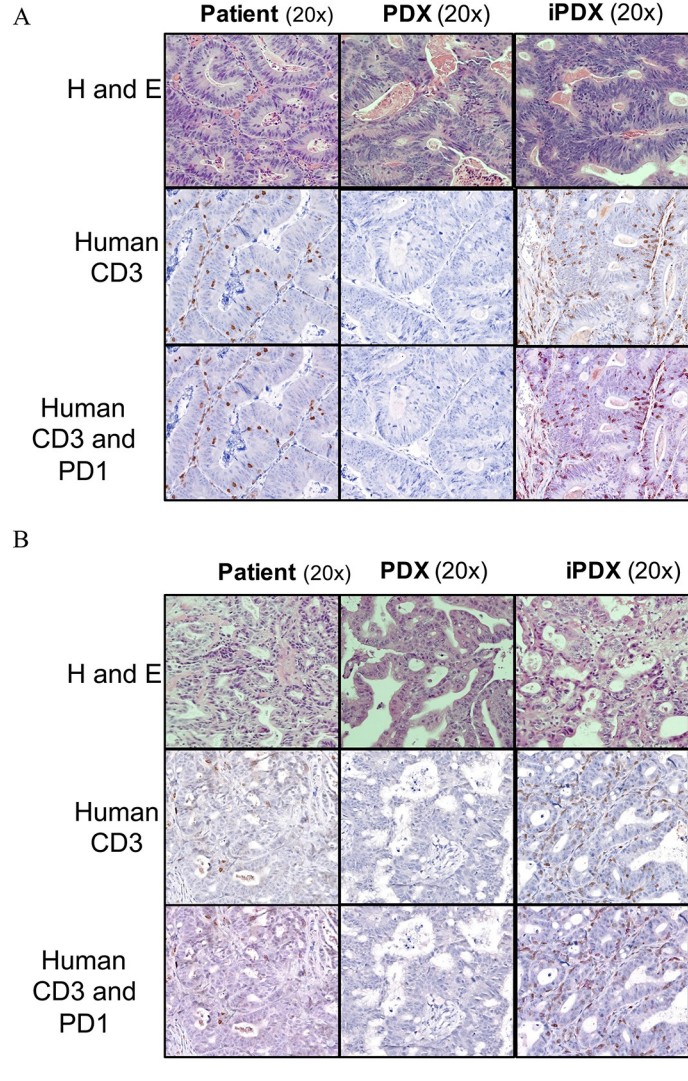

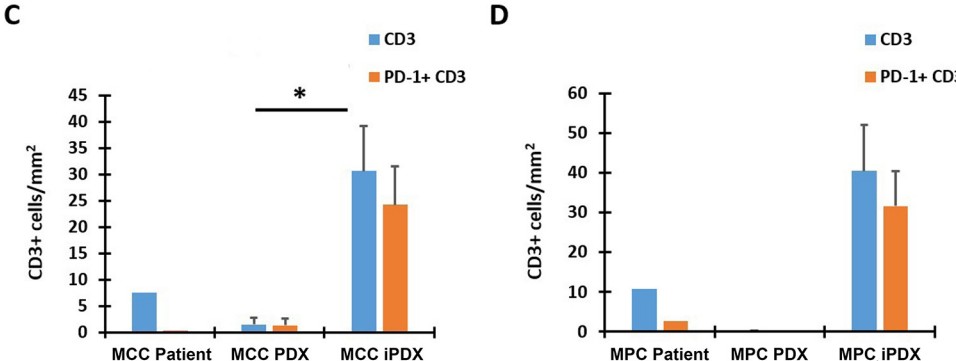

**Fig 2. Comparison of histopathologic features of the corresponding human tumor to the xenograft models.**
Histopathological evaluation of **(A)** MCC and **(B)** MPC models by Hematoxylin and Eosin staining (top row), human CD3 (middle row), and PD-1 and CD3 dual staining (bottom row) comparing the corresponding human tumor, PDX and iPDX tumors at 20X magnification. **(C)** and **(D)** Quantification of CD3 and PD-1 expression in the human cancer, MCC and MPC models. Data from samples collected at the end of the experiment at 7–8 weeks. Values represent mean of three animals per group ± SD. *p < 0.05.

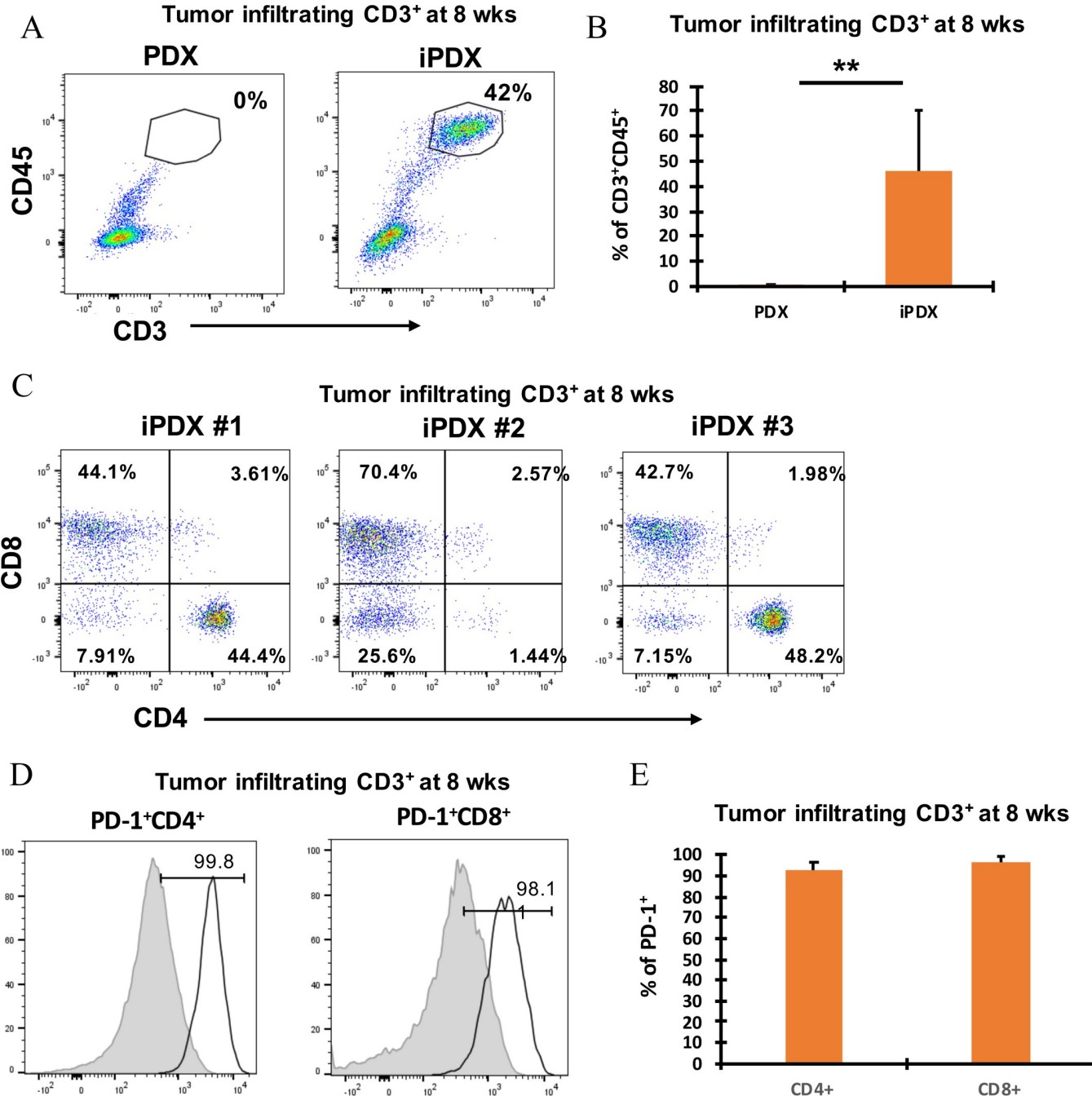

**Fig 3. Identification and phenotypic characterization of MCC iPDX tumor derived human CD3+ T cells. (A)** Representative flow cytometry dot plots showing CD3+CD45+ tumor infiltrating lymphocytes (TILs) (top left); **(B)** Bar graph of average values (top right). Data from samples collected at 8 weeks. (C) CD4+ and CD8+ TILs from 3 individual MCC iPDX mouse tumors (bottom panel). Data from samples collected at 8 weeks. **(D)** PD-1+CD4+ TILs (left) and PD-1+CD8+ TILs (middle) from a representative MCC iPDX mouse tumor. T lymphocytes from non-cancer human donor PB obtained from phlebotomy lab (grey histogram, left) and TILs from iPDX tumor (empty histogram, right). **(E)** Bar graph showing average percentage of PD-1+CD4+ and CD8+ T lymphocytes from 3 individual MCC iPDX mouse tumors. Data from samples collected at 8 weeks. Values represent mean of three animals per group ± SD. **p < 0.005.

### PBMC infusion was associated with reduced xenograft tumor growth

For the MCC and MPC models, the experiments ended on Day 50 and 39 respectively due to the increase in size of the control xenografts from the PDX (no PBMC infusion). Compared with the PDX, iPDX tumors demonstrated significantly less growth over time with a 61.5 ± 27.3% and 70.74 ± 5.1% observed reduction in growth at the end of the experiment for the MCC and MPC iPDX respectively (Fig 4A and 4C; p<0.014). As a secondary measure of tumor burden, the MCC iPDX xenograft average tumor weight was 62.3 ± 14.8 % less than the PDX xenograft at the time of harvest (Fig 4B). Similarly, the MPC iPDX mice demonstrated a 56.7 ± 6.1% decrease in average tumor weight compared with the PDX tumors (Fig 4D).

## Discussion

The lack of effective research platforms that accurately model the human tumor immune microenvironment remains a barrier to advancing immunotherapy experimental research. As

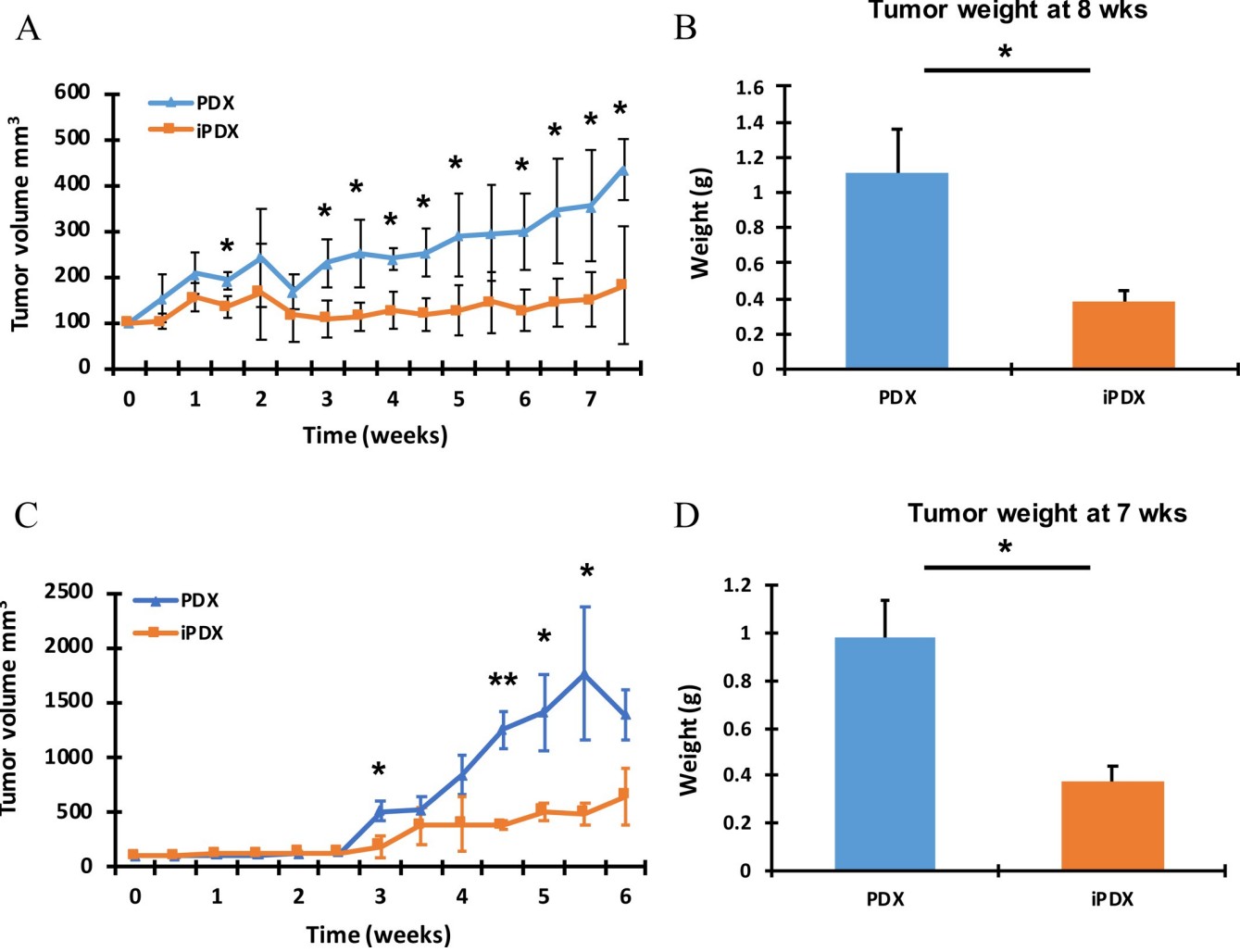

**Fig 4. iPDX demonstrate reduced tumor growth compared with the corresponding PDX.** Tumor **(A)** Average tumor volume (mm3) measured over time in the MCC model. **(B)** Final tumor weight (g) average in the MCC model. Data from samples collected at 8 weeks. **(C)** Average tumor volume (mm3) measured over time in the MPC model. **(D)** Final tumor weight (g) average in the MPC model. Data from samples collected at 7 weeks. Values represent mean of three animals per group ± SD. *p < 0.05. **p < 0.005.

Table 1. Features of current state-of-the-art mouse models.

| Mouse model | Advantages | Disadvantages |
|---|---|---|
| Mouse syngeneic PDAC cell lines [5] | • Intact immune system<br>• Tumors require implantation | • Tumors lack heterogeneity<br>• Non-human tumor-immune interaction |
| Genetic engineered mouse models [6] | • Elegantly models tumor progression<br>• Intact immune system | • Tumors lack heterogeneity<br>• Non-human tumor-immune interaction |
| HSPC mouse Models [16–19] | • Patient immune samples not required<br>• Can model multi-cellular immune system | • Tumor MHC1 mismatch<br>• T cells educated on mouse thymus |
| Bone marrow HSPC PDX models [16,19] | • Autologous immune source with MHC1 compatibility<br>• Can model multi-cellular immune system | • T cells educated on mouse thymus<br>• Low cellularity of immune samples<br>• Requires invasive procedure |
| PBMC- mouse model[12] | • Autologous immune source with MHC1 compatibility<br>• Ease of access to immune cells | • Xenogeneic graft vs. host disease<br>• Low % of human tumor-reactive T cells |
| TIL-PDX mouse Model [15,21] | • Autologous immune source with MHC1 compatibility<br>• High % of human tumor-reactive T cells<br>• No xenogeneic graft vs. host disease | • Requires access to fresh tumor<br>• Lacks standardization and validation<br>• Several weeks of culture for TIL expansion |
| Xenomimetic model (X mouse model) [29] | • Patient derived tumor neoantigen-specific T cells used<br>• Tumor xenografts expressing patient-derived tumor neoantigen peptides<br>• Fluorescence based imaging for detection of tumor burden | • Lacks architectural, genomic and morphological heterogeneity<br>• Single tumor cell line derived model |

the need and appeal for translational models to study immunotherapy grows, a variety of humanized platforms are being reported in the literature in an increasing fashion [15,16,19–21,28,29]. Humanized patient-derived xenograft mouse models represent an appealing pre-clinical strategy, but optimal model systems and standardized experimental parameters have not been defined.

Options for immune cell source include patient derived PBMCs, TILs, and HSPCs. Each strategy has unique advantages and disadvantages as highlighted in Table 1.

To obtain adequate TILs, immune cells require fresh tumors and then culturing for weeks to obtain an adequate cell number for experimentation [30,31]. HSPC models require a bone marrow biopsy to obtain autologous cells limiting the feasibility of this humanizing approach for solid cancers such as PDAC [16,19]. Thus, we elected to develop an autologous humanized PDX model using patient matched PBMCs and matched tumor samples from PDAC and CRC patients. A major advantage of this approach is that no additional procedures were required to obtain the necessary human autologous tissue samples. Our goal was to define experimental conditions to test immunotherapy and compare mouse model parameters to the human tumor for validation.

Autologous PBMC-based models are confounded by xGVHD which occurs primarily due to human T cell reactivity against mouse MHC molecules [22]. Parameters such as number of PBMCs infused and the use of sublethal radiation can impact xGVHD development [32,33]. Optimal parameters for PBMC-based systems have not been established even though multiple reports have used PBMC models to test immunotherapy [34–41]. We initially developed a pro-tocol with the goal of limiting xGVHD by infusing a low dose of PBMC and omitting whole-body radiation. Using these parameters, we observed only minimal signs of xGVHD for up to 8 weeks. In the tumor models, we infused human PBMCs after tumor xenograft implantation and growth to a volume of 100mm$^3$ to minimizes the impact of xGVHD on the experimental time course. This protocol for development of iPDX mice led to engraftment of human T cells in the peripheral blood and in the xenograft tumors although the degree of engraftment and change in cell populations over time was variable both within and between the models. The population of human T cells identified in the iPDX based on human CD3 staining was

significantly higher than detected in the corresponding human cancer. In addition, both CD4 + and CD8+ subsets were detected in the iPDX xenografts with elevated levels of PD-1 expression that was much higher than observed in the human tumors. The high levels of PD-1 expression observed in both the peripheral and tumoral T cells is concerning and could indicate non-specific activation of T cells likely due to xGVHD.

Although we successfully established human xenografts with infiltration of human T cells, our initial experience highlighted significant barriers with a humanized tumor xenograft model based on the use of autologous PBMCs. Notably, we encountered significant intra- and inter-model variability in the engrafted population of total human T cells (CD3+ cells) including both CD4+ and CD8+ subsets and tumor-infiltrating populations. The high variability in these parameters will need to be accounted for in future experimental models using this approach. We ultimately concluded that the intra- and inter-model variability is a critical limitation towards developing a reproducible experimental system that can rigorously test immunotherapy strategies. Using larger sample sizes and selecting mouse models based on engraftment might be an option to overcome the variability. However, a challenge with expanding the sample size is that the autologous human PBMCs represent a limited non-renewable resource that will require a higher blood volume or multiple blood draws. In addition, the variable T cell expansion rates over time point to the need for monitoring T cell populations over the course of an experiment.

Humanized PDX models remain an important opportunity to evaluate the interaction between the human immune system and a heterogeneous tumor model. Our work highlights this potential, as well as identifies critical limitations of an autologous PBMC-based humanized PDX approach. Prior studies using PBMC based platforms have typically used allogeneic human cancer cells, although a few investigations have used a limited number of patient-derived samples [36–38,42]. Model variability has not been thoroughly explored and attempts to validate based on patient parameters has not been investigated. Our comparison with the human patient tumor revealed important observations. We show that the iPDX maintains the essential heterogeneity of the matched human tumor. However, the xenograft had a much higher population of CD3+ cells that expressed high PD-1 relative to the patient tumor. These essential differences in the immune microenvironment are likely related to xGVHD as well as the lack of suppressive immune populations [43]. These inconsistencies may lead to experimental therapy results that cannot be reproduced in patients. The systemic xGVHD immune response observed in the iPDX mice should be considered as a confounding factor that can impact tumor growth and/ or attempts to model immunotherapy response.

Recognizing critical barriers to PBMC-based models, investigators have attempted to develop mouse models that can mitigate xGVHD. One appealing opportunity is to use NSG-($K^bD^b$) null mice that lack MHC1, which will likely further decrease cross-reactivity to extend the window for tumor development and therapy treatment before the onset of xGVHD [44]. Alternatively, NSG-HLA-A2.1 transgenic mice that express A2 MHC1 complexes could be used for HLA-A2$^+$ patient samples [45]. This strategy would enable human functional T-cell memory formation due to the expression of MHC class 1 on APCs but would restrict models to HLA-A2 patients (~50%). To support additional cell populations, transgenic mice have been developed and investigated as cancer models. The MISTRG model was created in immunocompromised mice to express human GM-CSF, CSF1, IL3 and thrombopoietin. This model was used to explore melanoma tumor biology and demonstrated an increased M2 macrophage tumor infiltration [19]. As experience grows with transgenic mice, the use of PBMCs as a patient-derived immune cell source for translational research will likely expand.

One additional concern with patient-derived PBMC-based cancer models is the low frequency of tumor reactive T cells. In contrast to PBMCs, TILs are highly enriched for tumor-

reactive T cells and may be a better source of cells that can reproduce the human tumor immune cell interaction [31]. A recent landmark report developed a model of adoptive cellular therapy using autologous IL-2 cultures of human tumor infiltrating lymphocytes infused into mice with matched melanoma xenografts [15]. A similar model was developed for ovarian cancer that incorporated an orthotopic intra-abdominal xenograft with intravenous infused TILs to study the efficacy of anti-PD-1 therapy [21]. Both reports cultured TILs *ex vivo* for weeks with IL-2 to establish adequate populations for the mouse models. A concern with this approach is that culturing tumor infiltrating T cells for extended periods may irreversibly change the biology of the associated T cells in a way that does not replicate the unmanipulated lymphocytes associated with a human tumor. Very recently, a novel xenograft platform utilizing patient-derived tumor neoantigen-specific T cells and tumor xenografts established from a melanoma cell line (DM6) expressing melanoma patient-derived tumor neoantigen peptides, as well as green fluorescent protein was described [29]. This strategy is an additional approach to improve the modeling of the human tumor-immune cell interaction. However, this single tumor cell line derived xenograft model lacks architectural, genomic, morphological heterogeneity of the human tumor. As experience with humanized models expands, investigators will be able to tailor the model strategies to the needs of the experiments.

## Conclusions

In conclusion, we effectively demonstrated that infusion of autologous human PBMCs can infiltrate patient-derived xenografts. Both human CD4+ T cells and CD8+ T cells were identified in the blood and xenograft tumors of the iPDX mice. Furthermore, iPDX maintained the histological features of the corresponding human tumors including the presence of CD3+ T cells suggesting that the iPDX model can recapitulate essential features of the complex human tumor. However, our results highlight the significant inter- and intra-model variability in lymphocyte engraftment and tumor immune cell infiltration that may limit future experimental applications. Future studies are ongoing to compare alternative sources of T cells and to assess tumor reactivity of tumor-engrafted T cells. Humanizing cancer models hold potential to fast-forward experimental immuno-therapy, although more work is needed to define and validate the optimal approach.

## Supporting information

**S1 Fig. Development of a humanized mouse model (iPDX).** Schematic showing the workflow and the time frame of humanized Pancreatic and Colorectal cancer iPDX mouse model establishment and analysis of T cell engraftment.
(PDF)

**S2 Fig. PBMC demonstrate limited signs of GVHD for up to 8 weeks in immuno-compromised mice.** NSG mice (6-8-week-old) were injected with 5 million hPBMCs (i.v) and GVHD was documented over a period of 12 weeks. **(A)** GVHD scores (n = 9). **(B)** Average weight change and GVHD scores (n = 9). **(C)** Weight change in individual mice (n = 9).
(PDF)

**S3 Fig. Effective engraftment of human PBMC derived T cells in MPC iPDX mouse peripheral blood.** (A) Representative flow cytometry plots are shown for MPC model CD3$^+$CD45$^+$ T lymphocytes from PDX mice (left), iPDX mice (middle). Data from samples collected at 2 weeks. (B) Bar graph of the average % of CD3$^+$CD45$^+$ cells (right). (C) Flow cytometry plots for CD4$^+$ and CD8$^+$ T lymphocytes from all iPDX mice (n = 3). Data from samples collected at 2 weeks. (D) The change in the average % of human T lymphocytes from

early (2 week) to late (7 weeks) engraftment are shown for MPC CD3$^+$CD45$^+$ T lymphocytes (left), CD8$^+$ T lymphocytes (middle) and CD4$^+$ T lymphocytes (right). (E) PD-1$^+$CD4$^+$ T lymphocytes (left) PD-1$^+$CD8$^+$ T lymphocytes (right). T lymphocytes from healthy human donor blood (grey histogram, left) and iPDX blood (empty histogram, right). (F) Bar graph showing average percentage of PD-1$^+$CD4$^+$ and CD8$^+$ T lymphocytes. Data from samples collected at 7 weeks. The values represent the percentage of human CD3, CD4 and CD8 population in iPDX mice peripheral blood. Representative averages of per group (n = 2) are shown. $^*$P<0.05 and $^{**}$P < 0.005.
(PDF)

**S4 Fig. Identification and phenotypic characterization of pancreatic iPDX tumor derived human CD3$^+$ T cells.** (A) Representative flow cytometry dot plots showing CD3$^+$CD45$^+$ tumor infiltrating lymphocytes (TILs) (top left); (B) Bar graph of average values (top right). Data from samples collected at 7 weeks. (C) CD4$^+$ and CD8$^+$ TILs from 2 individual MPC iPDX mouse tumors (bottom panel). (D) PD-1$^+$CD4$^+$ TILs (left) and PD-1$^+$CD8$^+$ TILs (middle) from a representative MPC iPDX mouse tumor. T lymphocytes from non-cancer human donor PB obtained from phlebotomy lab (grey histogram, left) and TILs from iPDX tumor (empty histogram, right). (E) Bar graph showing average percentage of PD-1$^+$CD4$^+$ and CD8$^+$ T lymphocytes from 2 individual MPC iPDX mouse tumors. Data from samples collected at 7 weeks. Values represent mean of two animals per group ± SD. $^{**}$p < 0.005.
(PDF)

**S1 File. Raw data.** Raw data (such as values behind the means, standard deviations and other measures reported) required to replicate the study.
(XLSX)

## Acknowledgments

Marzena Swiderska-Syn provided technical assistance for this research work.

## Author Contributions

**Conceptualization:** Harinarayanan Janakiraman, Ernest Ramsay Camp.

**Data curation:** Harinarayanan Janakiraman, Scott A. Becker.

**Formal analysis:** Harinarayanan Janakiraman, Scott A. Becker, Mark P. Rubinstein.

**Funding acquisition:** Ernest Ramsay Camp.

**Investigation:** Harinarayanan Janakiraman.

**Methodology:** Harinarayanan Janakiraman, Scott A. Becker, Alexandra Bradshaw.

**Project administration:** Harinarayanan Janakiraman.

**Resources:** Mark P. Rubinstein, Ernest Ramsay Camp.

**Supervision:** Ernest Ramsay Camp.

**Validation:** Harinarayanan Janakiraman.

**Writing – original draft:** Harinarayanan Janakiraman, Ernest Ramsay Camp.

**Writing – review & editing:** Harinarayanan Janakiraman, Mark P. Rubinstein, Ernest Ramsay Camp.

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
