## [Decision Letter · Decision Letter 0]

14 Feb 2022

PONE-D-21-34701Critical evaluation of an autologous peripheral blood mononuclear cell-based humanized cancer modelPLOS ONE

Dear Dr. Janakiraman,

Thank you for submitting your manuscript to PLOS ONE. After careful consideration, we feel that it has merit but does not fully meet PLOS ONE’s publication criteria as it currently stands. Therefore, we invite you to submit a revised version of the manuscript that addresses the points raised during the review process.

We look forward to receiving your revised manuscript.

Kind regards,

Anilkumar Gopalakrishnapillai

Academic Editor

PLOS ONE

Journal Requirements:

Supported in part by the Translational Science Shared Resource, Hollings Cancer Center, Medical University of South Carolina (P30 CA138313). Supported in part by the Biorepository & Tissue Analysis Shared Resource, Hollings Cancer Center, Medical University of South Carolina.

Additional Editor Comments:

I would like authors go through the comments of the reviewers and include all the responses in the revised manuscript. I specifically would like to see authors including an immunotherapy modality in this model as a proof of principle to validate the work.

Reviewers' comments:

Reviewer's Responses to Questions

**Comments to the Author**

1. Is the manuscript technically sound, and do the data support the conclusions?

Reviewer #1: Yes

Reviewer #2: Yes

2. Has the statistical analysis been performed appropriately and rigorously? 

Reviewer #1: Yes

Reviewer #2: Yes

3. Have the authors made all data underlying the findings in their manuscript fully available?

Reviewer #1: Yes

Reviewer #2: Yes

4. Is the manuscript presented in an intelligible fashion and written in standard English?

Reviewer #1: Yes

Reviewer #2: Yes

5. Review Comments to the Author

Reviewer #1: The article entitled "Critical evaluation of an autologous peripheral blood mononuclear cell-based humanized cancer model". Authors have established an autologous humanized PDX model for reproducible immunotherapy investigation. They have aimed to determine the technical aspects required to establish an autologous humanized cancer model system and demonstrated the ability to successfully engraft patient PBMCs that infiltrate the xenograft.

The paper is well-written. Authors have highlighted the potential of using PDX models to evaluate the interaction between the human immune system and a heterogeneous tumour model, in addition to the identification of the limitations of an autologous PBMC-based humanized PDX approach. The article falls within the scope of PLOS ONE and is recommended to be considered for publication.

Reviewer #2: This is a well written manuscript describing the development of an autologous PBMC derived PDX models in two different types of solid tumors for preclinical evaluation of immunotherapy. Comparison between the patient and mouse model is a bonus.

Major

Use of an immunotherapy modality in this model at least as proof-of-principle is lacking.

Minor

Fig. 3F How can the % of both CD4 and CD8 T cells be close to 80%? It is confusing as these markers are mutually exclusive.

Figures 1 and 2 should be put in the supplement.

6. PLOS authors have the option to publish the peer review history of their article (what does this mean?). If published, this will include your full peer review and any attached files.

Reviewer #1: No

Reviewer #2: No

---

## [Author Response · Author response to Decision Letter 0]

8 Mar 2022

Editor Comments: 

1. specifically would like to see authors including an immunotherapy modality in this model as a proof of principle to validate the work. 

We agree that it would be ideal to test an immunotherapy modality in our model to broaden our conclusions. In fact, this was the overarching goal as we stated in the introduction section. However, for the reasons outlined below, we concluded that a patient-derived autologous PBMC method for humanization may not be the optimal strategy to study cancer immunotherapy in a rigorous and reproducible manner. Thus, this manuscript is focused on our model development approach and the barriers that we encountered in optimizing experimental conditions. 

 Our goal was to develop a patient-derived humanized cancer mouse model based on the widely cited reference (Sanmamed et al. Cancer Research, 2015) that used a single gastric cancer patient tumor and PBMCs to study the use of Nivolumab and Urelumab. In retrospect, this elegant experiment only reported using a single patient sample and we could not find that it was repeated with the same patient samples or with another patient tissue samples. Our attempt to reproduce the published humanized model for immunotherapy testing with both colorectal cancer and pancreatic cancer tissue samples identified critical barriers that challenged the feasibility of the approach including: 

• Early evidence of xGVHD in the xenograft prior to clinical xGVHD in the mouse.

• Significant intra- and inter model immune cell engraftment variability.

 These unexpected observations ultimately led us to conclude that a PBMC-based humanized model would not result in a reproducible model system that reflects the human condition for testing cancer immunotherapy rigorously. The degree of variability was particularly concerning and challenged the feasibility of the model for immunotherapy testing. In addition, the limited PBMC supply obtained from cancer patients decreased our ability to repeatedly test models. To further emphasize the valuable points our investigations uncovered, we have revised the manuscript. We have now emphasized these barriers in the manuscript conclusions. On page 24 line 373, we revised the manuscript as “We ultimately concluded that the intra- and inter-model variability is a critical limitation towards developing a reproducible experimental system that can rigorously test immunotherapy strategies. Using larger sample sizes and selecting mouse models based on engraftment might be an option to overcome the variability.” In the abstract, we revised the final statements as “In summary, the iPDX models can reproduce key features of the corresponding human tumor. However, the observed variability in T cell engraftment and high PD-1 expression are important considerations that need to be addressed in order to develop a reproducible model system.”

 While we were not able to achieve the original goal of our studies with the resources available for this effort, we believe that our manuscript will be very informative to future investigators who attempt a patient-derived autologous PBMC-humanized approach. On the surface, the model system is very appealing due to the relative ease of obtaining the needed patient samples. The issues we outline can provide future investigators with a realistic appreciation of the limitations and we expect the manuscript will serve as a vital reference for future investigators seeking to develop humanized mouse models for assessing immunotherapy.

Reviewer’s comments and response.

Reviewer #1:

Thank you for the comments that you believe our paper is “well written” and that you recommend for publication.

Reviewer #2

1. Use of an immunotherapy modality in this model at least as proof-of-principle is lacking. 

 Please refer to our comments to the Editor above. We completely agree and we were sincerely disappointed in the significant limitations that we encountered that were previously not reported or discussed. We hope that you appreciate our effort to reproducibly create an effective model and that our rigorous testing demonstrated critical limitations that we decided were important to report in order to inform future investigators.

2. Fig. 3F How can the % of both CD4 and CD8 T cells be close to 80%? It is confusing as these markers are mutually exclusive. Thank you for your comment. We intended to demonstrate the % of cell populations that were PD-1 positive in the xenografts. The total CD4 and CD8 T cell populations are demonstrated in 3D. To make figure 3F less confusing, we have bolded and enlarged the Y-axis title to make this clearer that it is only the %PD-1 positive cell populations in the analysis. 

3. Figures 1 and 2 should be put in the supplement. We agree and now have moved these figures to the supplemental section and revised the figure legend accordingly.

Other requirements:

4. We have revised the Methods section to now include (1) methods of sacrifice (line 126), (2) methods of anesthesia and/or analgesia (line 130), and (3) efforts to alleviate suffering (line 137-138).

5. Funding statement has been updated and now includes the statement “There was no additional external funding received for this study.”

6. Funding Information’ and ‘Financial Disclosure’ sections now match.

---

## [Editor Report · Decision Letter 1]

15 Jul 2022

PONE-D-21-34701R1

Critical evaluation of an autologous peripheral blood mononuclear cell-based humanized cancer model

PLOS ONE

Dear Dr. Janakiraman,

Thank you for submitting your manuscript to PLOS ONE. After careful consideration, we feel that it has merit but does not fully meet PLOS ONE’s publication criteria as it currently stands. Therefore, we invite you to submit a revised version of the manuscript that addresses the points raised during the review process.

We look forward to receiving your revised manuscript.

Kind regards,

Anilkumar Gopalakrishnapillai

Academic Editor

PLOS ONE

Additional Editor Comments:

One of the major comments of the reviewer was to test the feasibility of this model for immunotherapy purpose. Please address this major comment along with others.
---

## [Author Response · Author response to Decision Letter 1]

25 Jul 2022

Editor Comments: 

1. specifically would like to see authors including an immunotherapy modality in this model as a proof of principle to validate the work. 

We agree that it would be ideal to test an immunotherapy modality in our model to broaden our conclusions. In fact, this was the overarching goal as we stated in the introduction section. However, for the reasons outlined below, we concluded that a patient-derived autologous PBMC method for humanization may not be the optimal strategy to study cancer immunotherapy in a rigorous and reproducible manner. Thus, this manuscript is focused on our model development approach and the barriers that we encountered in optimizing experimental conditions. 

 Our goal was to develop a patient-derived humanized cancer mouse model based on the widely cited reference (Sanmamed et al. Cancer Research, 2015) that used a single gastric cancer patient tumor and PBMCs to study the use of Nivolumab and Urelumab. In retrospect, this elegant experiment only reported using a single patient sample and we could not find that it was repeated with the same patient samples or with another patient tissue samples. Our attempt to reproduce the published humanized model for immunotherapy testing with both colorectal cancer and pancreatic cancer tissue samples identified critical barriers that challenged the feasibility of the approach including: 

• Early evidence of xGVHD in the xenograft prior to clinical xGVHD in the mouse.

• Significant intra- and inter model immune cell engraftment variability.

 These unexpected observations ultimately led us to conclude that a PBMC-based humanized model would not result in a reproducible model system that reflects the human condition for testing cancer immunotherapy rigorously. The degree of variability was particularly concerning and challenged the feasibility of the model for immunotherapy testing. In addition, the limited PBMC supply obtained from cancer patients decreased our ability to repeatedly test models. To further emphasize the valuable points our investigations uncovered, we have revised the manuscript. We have now emphasized these barriers in the manuscript conclusions. On page 24 line 373, we revised the manuscript as “We ultimately concluded that the intra- and inter-model variability is a critical limitation towards developing a reproducible experimental system that can rigorously test immunotherapy strategies. Using larger sample sizes and selecting mouse models based on engraftment might be an option to overcome the variability.” In the abstract, we revised the final statements as “In summary, the iPDX models can reproduce key features of the corresponding human tumor. However, the observed variability in T cell engraftment and high PD-1 expression are important considerations that need to be addressed in order to develop a reproducible model system.”

 While we were not able to achieve the original goal of our studies with the resources available for this effort, we believe that our manuscript will be very informative to future investigators who attempt a patient-derived autologous PBMC-humanized approach. On the surface, the model system is very appealing due to the relative ease of obtaining the needed patient samples. The issues we outline can provide future investigators with a realistic appreciation of the limitations and we expect the manuscript will serve as a vital reference for future investigators seeking to develop humanized mouse models for assessing immunotherapy.

Reviewer’s comments and response.

Reviewer #1:

Thank you for the comments that you believe our paper is “well written” and that you recommend for publication.

Reviewer #2

1. Use of an immunotherapy modality in this model at least as proof-of-principle is lacking. 

 Please refer to our comments to the Editor above. We completely agree and we were sincerely disappointed in the significant limitations that we encountered that were previously not reported or discussed. We hope that you appreciate our effort to reproducibly create an effective model and that our rigorous testing demonstrated critical limitations that we decided were important to report in order to inform future investigators.

2. Fig. 3F How can the % of both CD4 and CD8 T cells be close to 80%? It is confusing as these markers are mutually exclusive. Thank you for your comment. We intended to demonstrate the % of cell populations that were PD-1 positive in the xenografts. The total CD4 and CD8 T cell populations are demonstrated in 3D. To make figure 3F less confusing, we have bolded and enlarged the Y-axis title to make this clearer that it is only the %PD-1 positive cell populations in the analysis. 

3. Figures 1 and 2 should be put in the supplement. We agree and now have moved these figures to the supplemental section and revised the figure legend accordingly.

Other requirements:

4. We have revised the Methods section to now include (1) methods of sacrifice (line 126), (2) methods of anesthesia and/or analgesia (line 130), and (3) efforts to alleviate suffering (line 137-138).

5. Funding statement has been updated and now includes the statement “There was no additional external funding received for this study.”

6. Funding Information’ and ‘Financial Disclosure’ sections now match.

Overall, we are grateful for the possible opportunity to publish our manuscript in PLOS ONE. We believe that our critical evaluation of a PBMC humanized cancer model system will provide important reference points for future investigators. Please do not hesitate to contact me should any questions arise. Thank you for your consideration in this matter.

---

## [Editor Report · Decision Letter 2]

3 Aug 2022

Critical evaluation of an autologous peripheral blood mononuclear cell-based humanized cancer model

PONE-D-21-34701R2

Dear Dr. Janakiraman,

We’re pleased to inform you that your manuscript has been judged scientifically suitable for publication and will be formally accepted for publication once it meets all outstanding technical requirements.

Kind regards,

Anilkumar Gopalakrishnapillai

Academic Editor

PLOS ONE

---

## [Editor Report · Acceptance letter]

2 Sep 2022

PONE-D-21-34701R2 

Critical evaluation of an autologous peripheral blood mononuclear cell-based humanized cancer model 

Dear Dr. Janakiraman:

I'm pleased to inform you that your manuscript has been deemed suitable for publication in PLOS ONE. Congratulations! Your manuscript is now with our production department. 

Kind regards, 

on behalf of

Dr. Anilkumar Gopalakrishnapillai 

Academic Editor

PLOS ONE